# Antiproliferative Activity of Krukovine by Regulating Transmembrane Protein 139 (TMEM139) in Oxaliplatin-Resistant Pancreatic Cancer Cells

**DOI:** 10.3390/cancers15092642

**Published:** 2023-05-07

**Authors:** Jee-Hyung Lee, Sang-Hyub Lee, Sang-Kook Lee, Jin-Ho Choi, Seohyun Lim, Min-Song Kim, Kyung-Min Lee, Min-Woo Lee, Ja-Lok Ku, Dae-Hyun Kim, In-Rae Cho, Woo-Hyun Paik, Ji-Kon Ryu, Yong-Tae Kim

**Affiliations:** 1Department of Internal Medicine and Liver Research Institute, Seoul National University Hospital, Seoul National University College of Medicine, Seoul 03080, Republic of Korea; granule@snu.ac.kr (J.-H.L.); 1108lsh@naver.com (S.L.); msong0701@gmail.com (M.-S.K.); km601@naver.com (K.-M.L.); snuhimchief@snu.ac.kr (M.-W.L.); inrae0428@snu.ac.kr (I.-R.C.); iatrus@snu.ac.kr (W.-H.P.); jkryu@snu.ac.kr (J.-K.R.); yongtkim@snu.ac.kr (Y.-T.K.); 2Natural Products Research Institute, Seoul National University College of Pharmacy, Seoul 08826, Republic of Korea; 3Department of Medicine, Samsung Medical Center, Sungkyunkwan University School of Medicine, Seoul 03181, Republic of Korea; jinhchoi@snu.ac.kr; 4Department of Biomedical Sciences, Korean Cell Line Bank, Laboratory of Cell Biology and Cancer Research Institute, Seoul National University College of Medicine, Seoul 03080, Republic of Korea; kujalok@snu.ac.kr; 5Dxome Co., Ltd., Seongnam 331, Republic of Korea; dkim@dxome.com

**Keywords:** pancreatic cancer, krukovine, *KRAS*, transmembrane protein 139 (TMEM139), metastasis, patient-derived pancreatic cancer organoid (PDPCO), oxaliplatin, combination

## Abstract

**Simple Summary:**

This study investigated the antiproliferative activity of Krukovine (KV) in oxaliplatin-resistant pancreatic cancer cells and explored the mechanism of action. KV suppresses tumor progression via the downregulation of the Erk-RPS6K-TMEM139 signaling pathway in oxaliplatin-resistant pancreatic cancer cells. Transmembrane protein 139 (TMEM139) has been identified as a novel oncogene and its overexpression has been associated with various types of cancer, including pancreatic cancer. In this study, KV significantly downregulated TMEM139 expression in oxaliplatin-resistant pancreatic cancer cells. The downregulation of TMEM139 by KV may contribute to its antiproliferative activity and could be a potential target for future therapeutic interventions. KV could be a potential therapeutic agent for the treatment of pancreatic cancer, particularly for patients with *Kras* mutations and oxaliplatin-resistant tumors.

**Abstract:**

Krukovine (KV) is an alkaloid isolated from the bark of *Abuta grandifolia* (Mart.) Sandw. (Menispermaceae) with anticancer potential in some cancers with *KRAS* mutations. In this study, we explored the anticancer efficacy and mechanism of KV in oxaliplatin-resistant pancreatic cancer cells and patient-derived pancreatic cancer organoids (PDPCOs) with *KRAS* mutation. After treatment with KV, mRNA and protein levels were determined by RNA-seq and Western blotting, respectively. Cell proliferation, migration, and invasion were measured by MTT, scratch wound healing assay, and transwell analysis, respectively. Patient-derived pancreatic cancer organoids (PDPCOs) with *KRAS* mutations were treated with KV, oxaliplatin (OXA), and a combination of KV and OXA. KV suppresses tumor progression via the downregulation of the Erk-RPS6K-TMEM139 and PI3K-Akt-mTOR pathways in oxaliplatin-resistant AsPC-1 cells. Furthermore, KV showed an antiproliferative effect in PDPCOs, and the combination of OXA and KV inhibited PDPCO growth more effectively than either drug alone.

## 1. Introduction

Pancreatic cancer is a malignant tumor that is continually deleterious throughout its progression with an average length of survival of only 6 months [1]. Chemotherapy, radiation therapy, and surgical operation are considered the most common and effective strategies for the treatment of early-stage cancers [2]. However, most pancreatic cancer cases are diagnosed at advanced stages [3]. Therefore, only 15–20% of patients with pancreatic cancer are eligible for surgical treatment, and over 80% undergo medicinal treatment [4].

Combination therapies such as FOLFIRINOX (leucovorin + 5-fluorouracil + oxaliplatin + irinotecan) and gemcitabine plus nab-paclitaxel are commonly used to treat patients with pancreatic cancer [5]. Although oxaliplatin in FOLFIRINOX is widely used as a first-line chemotherapy for pancreatic cancer patients [6], acquired resistance to oxaliplatin has increasingly become a problem. Therefore, there is an urgent need for a novel therapeutic strategy to overcome oxaliplatin resistance in pancreatic cancer patients.

Pancreatic ductal adenocarcinoma is an extremely aggressive disease with high metastatic potential. Most patients are diagnosed with metastatic disease which is the major cause of death in pancreatic cancer patients [7]. Although there have been several attempts to develop effective agents for the treatment of pancreatic cancer metastasis, the current therapeutic options are limited due to lower selectivity [8]. Therefore, novel strategies are urgently required for the efficient treatment of patients with metastatic pancreatic cancer.

The transmembrane (TMEM) protein family is related to the formation of metastases and the mechanisms leading to cancer cell dissemination such as migration and extra-cellular matrix remodeling [9]. Though the function of TMEMs remains largely unknown, a body of emerging evidence has revealed that TMEMs play a role in tumor occurrence and progression [9]. Recent studies have reported that TMEMs are involved in the regulation of cell proliferation, invasion, metastasis, and chemoresistance [10]. Moreover, several TMEMs were found to be correlated with the overall survival of patients and act as prognostic biomarkers in multiple tumors [10]. These findings strongly suggest that TMEM inhibitors may be effective antitumor agents in the treatment of metastatic pancreatic cancers.

In most pancreatic cancer patients, proto-oncogenes such as *KRAS* (Kirsten rat sarcoma viral oncogene homolog) are constitutionally activated, leading to uncontrolled cell proliferation, apoptotic resistance, and other oncogenic cascades [11,12]. *KRAS* mutation is the most frequent mutation (more than 90%) and the initiating genetic event for pancreatic ductal adenocarcinoma (PDAC), and it is found in primary tumors, metastatic tumors, and even in pancreatic intraepithelial neoplasia (PanIN), the earliest preneoplastic stage in PDAC progression [13]. Therefore, the *KRAS* signaling pathway is considered one of the most important targets to develop novel agents to treat pancreatic cancer [14]. In this study, we used *KRAS*-mutated pancreatic cancer cells to explore its underlying mechanism by KV treatment.

Given their low toxicity and high effectiveness, natural products have been studied and used worldwide as potential anticancer agents [15,16,17,18,19]. Krukovine (KV) is a bisbenzylisoquinoline alkaloid derived from *Abuta grandifolia* (Mart.) Sandw. (Menispermaceae) [13,20]. Bisbenzylisoquinoline alkaloid-bearing natural products have exhibited various pharmacological effects, such as anti-inflammation, antitumor, and antiviral activity [21]. In a previous study, KV showed an anticancer effect in *KRAS*-mutated lung cancer via the PI3K-AKT-mTOR and RAF-ERK signaling pathways [13]. These findings encouraged us to further examine KV activity in pancreatic cancers for targeting *KRAS*-mutated pancreatic cancer as a natural product-derived antitumor agent.

Organoids are stem cell-derived 3D model systems that resemble human organs, thus making it possible to mimic the architecture and physiology of human organs in remarkable detail. Organoids recently emerged as adequate models for predicting drug response, identifying cancer therapeutic targets, and detecting cancer patients [22,23,24,25,26]. The tumor microenvironment (TME) plays a critical role in tumorigenesis and resistance to therapy and is comprised of different components, including tumor cells, stromal cells, and immune cells [27]. The coculture of patient-derived cancer organoids with these cells enables one to mimic the TME. Therefore, many researchers have tried to make the TME through the organoid culture system [28]. Furthermore, patient-derived cancer organoids can represent the patient’s genetic characteristics, and thus may be an advanced tool compared to 2D cell lines and spheroids for predicting drug response in pancreatic cancer patients.

In this study, the antiproliferative and antimetastatic activities of KV, a bisbenzylisoquinoline alkaloid, were elucidated in *KRAS*-mutated oxaliplatin-resistant pancreatic cancer cells (oxaliplatin-resistant AsPC-1 cells). Specifically, the underlying molecular mechanisms of TMEM (TMEM139) against oxaliplatin-resistant AsPC-1 cells were investigated. Furthermore, organoid models were used to evaluate the effect of KV for pancreatic cancer patients.

## 2. Materials and Methods

### 2.1. Cell Cultures and Reagents

*KRAS*-mutated pancreatic cancer cell lines, AsPC-1 (c.35G > A), Panc-1 (c.35G > A), and MiaPaCa-2 (c.34G > T) were obtained from the ATCC (American Type Culture Collection). The cells were cultured in RPMI (or DMEM, Gibco, Thermo Fisher Scientific, Waltham, MA, USA) with 10% FBS and 1% streptomycin/penicillin at 37 °C in a 5% CO_2_ incubator. KV was purchased from Specs (Bleiswijkseweg, The Netherlands) and dissolved in dimethyl sulfoxide (DMSO). Oxaliplatin was purchased from Sigma-Aldrich (Shanghai, China). KV and oxaliplatin were dissolved in dimethyl sulfoxide (DMSO; Sigma, Saint Louis, MO, USA) and added to phosphate-buffered saline (PBS), used as a storage solution. The solution was then added into the cell culture medium at various concentrations. The final concentration of DMSO was <0.1% (*v*/*v*) in all experiments. The Annexin V/PI staining kit was produced by BD Biosciences (San Jose, CA, USA).

### 2.2. MTT Assay (Cell Viability Assay)

A 5-Diphenyltetrazolium bromide (MTT) (Sigma-Aldrich) assay was performed to measure cell proliferation in 2D cell lines. Pancreatic cancer cells (2 × 10^4^ cells/mL) were cultured in 96-well plates with different doses of KV or oxaliplatin. The cells were then treated with KV or oxaliplatin (from 0 to 100 μM) to generate curves and measure the half-maximal inhibitory concentration (IC_50_) values. Following drug treatment, 0.5 mg/mL MTT was added into each well at 24 and 48 h, and cells were further incubated for 4 h at room temperature (RT). The supernatants were then discarded, and colored formazan crystals were dissolved with 150 μL/well of DMSO. A microplate reader (Bio-Rad, Hercules, CA, USA) at 570 nm was used to analyze the OD values.

### 2.3. RNA Preparation, Library Preparation, and RNA-Seq

The oxaliplatin-resistant AsPC-1 cells treated with KV (IC_50_ concentration) were collected from three randomly selected plates. Total RNA was extracted from the oxaliplatin-resistant AsPC-1 cell plates treated with KV using TRIzol reagent (Invitrogen Life Technologies, Carlsbad, CA, USA) according to the manufacturer’s instructions. After synthesizing cDNA libraries, their quality was evaluated using an Agilent 2100 BioAnalyzer (Agilent, Santa Clara, CA, USA). The cDNA libraries were quantified using the KAPA Library Quantification Kit (Kapa Biosystems, Boston, MA, USA). After cluster amplification of the denatured templates, samples in flow cells were sequenced as paired-end polymers (2 × 100 bp) using the Illumina HiSeq2500 (Illumina, San Diego, CA, USA).

#### 2.3.1. Preprocessing of the RNA-Seq Data

Low-quality reads were filtered out according to the following criteria: reads containing >10% of skipped bases (marked as Ns), reads containing >40% of bases whose quality scores were <20, and reads whose average quality score was <20. The filtering process was performed using in-house scripts. The remaining reads were mapped onto the mouse reference genome (Ensembl, release 72) using the aligner software STAR version 2.3.0e. The gene expression levels were measured using Cufflinks version 2.1.1, using the gene annotation database of Ensembl, release 72. The noncoding gene regions were removed by means of the mask option. To improve the accuracy of the measurement, “multiread correction” and “frag bias-correct” options were used. All other options were set to the default values.

#### 2.3.2. Differential Transcriptome and Functional Analysis

For differential expression analysis, the data on gene-level counts were generated using HTSeq-count version 0.5.4p3. Using the resulting read count data, differentially expressed genes (DEGs) were identified using the R software package, TCC (Bioconductor open source project). The TCC package uses robust normalization strategies to compare tag count data. Normalization factors were calculated using the iterative DEGES/edgeR method. The *q*-value was calculated from the *p*-value using the *p*. adjust function in the R package and the default settings. DEGs were identified based on a *q*-value threshold of less than 0.05. K-means clustering was performed in the Bioinformatics Toolbox of MATLAB R2009a.

#### 2.3.3. Molecular Pathway and Functional Analysis

The DEG lists were analyzed using the Ingenuity Pathway Analysis (IPA) software (IPA, Ingenuity^®^ systems, Qiagen, CA, USA). IPA allows for the identification of network interactions and pathway interactions between genes, based on an extensive manually curated database of published gene interactions. We uploaded the genes with a *q*-value threshold of less than 0.05, and a fold change in expression of more than 1.5, after HFD, with or without GT supplementation, and the associated expression value from the RNA-seq data into IPA. GO (Gene Ontology) enrichment analysis was performed using the DAVID database to perform on, and the significantly enriched GO terms were obtained (*p* < 0.05).

### 2.4. Kaplan–Meier Plotter Analysis

The Kaplan–Meier Plotter (http://kmplot.com/analysis/ (accessed on 5 October 2022) analysis was used to evaluate the overall survival (OS) and relapse-free survival (RFS) rates of patients with pancreatic cancer. The auto-select best cutoff method was used to classify patients with pancreatic cancer. Hazard ratios (HRs) with 95% confidence intervals (CIs) and log rank *p*-values were also calculated.

### 2.5. Protein–Protein Interaction (PPI) Network Analysis

PPI network analysis is a search tool for the retrieval of interacting genes (STRING) (https://string-db.org (accessed on 5 October 2022) database, which integrates both known and predicted PPIs and can be applied to predict functional interactions of proteins. To seek potential interactions between DEGs according to RNA-seq results, the STRING tool was employed.

### 2.6. Western Blot

Cells and organoids were lysed with RIPA buffer containing protease inhibitors and phosphatase inhibitors (Sigma Aldrich, St Louis, MO, USA). Total cell lysates were prepared in 5× sample loading buffer (250 mM Tris-HCl (pH 6.8), 40% glycerol (80%, *v*/*v*), 8% sodium dodecyl sulfate (SDS), 2% β-mercaptoethanol, 0.1% bromophenol blue, 100 mM Dithiothreitol (1M); Cold Spring Harbor, NY, USA). The protein concentrations of samples were quantified using the bicinchoninic acid (BCA) method and a BCA Protein Assay Kit (Thermo Fisher Scientific, Waltham, MA, USA). Equal amounts of protein (20–30 µg) were separated by 6–13% SDS-polyacrylamide gel electrophoresis (PAGE) and transferred to polyvinylidene fluoride membranes (Millipore, Bedford, MA, USA). The membranes were blocked with 5% bovine serum albumin (Sigma-Aldrich). The antibodies for Western blot are as follows: primary antibodies against phospho-ERK (Thr202-Thy204), total-ERK, TMEM139, RPS6K, phospho-AKT (Ser473), phospho-PI3K, total-PI3K (p110α), phospho-mTOR (Ser2448), total-mTOR, N-cadherin, snail, slug, twist, phosphor-CRAF, total-CRAF, PARP, cleaved-PARP, and α-tubulin by CST (Cell Signaling Technology, United States). The primary antibody against total-AKT was produced by Santa Cruz Biotechnology (Santa Cruz, CA, USA). Fluorescein-conjugated goat anti-rabbit and -mouse secondary antibodies were produced by Santa Cruz Biotechnology (Santa Cruz, CA, USA). Blots were detected using a chemiluminescence detection kit (Pierce ECL Western blotting substrate, Thermo Scientific, Rockford, IL, USA).

### 2.7. Wound Healing Assay (Cell Migration Assay)

A scratch wound healing assay was used to detect cell migration. Oxaliplatin-resistant AsPC-1 cells were seeded into six-well plates (500 cells/well), respectively. After attachment overnight, the confluent monolayer (~70%) was scratched using a 200 μL pipette tip. The well was gently washed with medium to wash the detached cells and treated with 0, 12.5, 25, or 50 μM KV (in 10% FBS-supplemented RPMI medium). After growing for 48 h, the cells were washed with PBS. Cell migration (width) was measured at 0 and 48 h with an inverted microscope at 100× magnification.

### 2.8. Transwell Cell Invasion Assay

Twenty-four-well transwell membrane inserts (diameter: 6.5 mm, pore size: 8 µm; Corning, Tewksbury, MA, USA) were each coated with 10 µL of type I collagen (0.5 mg/mL, BD Biosciences, San Diego, CA, USA) and 20 µL of a 1:20 mixture of Matrigel (BD Biosciences) in PBS. Oxaliplatin-resistant AsPC-1 cells were seeded into six-well plates (2 × 10^4^ cells/mL). After treatment with different concentrations of 0, 12.5, 25, or 50 μM KV for 48 h, oxaliplatin-resistant AsPC-1 cells were harvested from each group, resuspended in serum-free medium, and plated (1 × 10^5^ cells/ chamber) in the upper chambers of the matrigel-coated transwell inserts. A medium (600 μL) containing 30% FBS was used as a chemoattractant in the lower chambers. After 48 h of incubation, the cells that had migrated to the outer surfaces of the lower chambers were fixed and stained with 0.5% crystal violet (20% methanol, 0.5% crystal violet, and 1% PFA in ddH_2_O) for 30 min. After the extra crystal violet was washed away with distilled water and dried off, the colonies were photographed under microscopy (inverted microscope; Olympus CKX41, Shibuya, Tokyo, Japan) using 40× magnification. Each experiment was performed in triplicate and counted. Representative images from three separate experiments are shown.

### 2.9. Organoid Medium

Patient-derived pancreatic cancer organoids (PDPCOs) were provided by Korean Cell Line Bank and cultured at 37 °C in 5% CO_2_/95% air in complete media; 50% (*v*/*v*) with Wnt-3A, R-spondin 1, and mNoggin conditioned medium, 50 ng/mL human epidermal growth factor, 100 ng/mL human Fibroblast Growth Factor 10, 10 mM Nicotinamide, 500nM A83-01, 1× B27 supplement, 1.25 mM N-acetylcysteine, 10% FBS, and 0.01 μM human Gastrin 1 in 50% DMEM culture medium (containing 10% fetal bovine serum (FBS) and 1% antibiotics). Recombinant Wnt-3A, Noggin, and R-spondin1 can be substituted with conditioned medium from L-WRN (ATCC^®^ CRL-3276TM) cell line. L-WRN cell line was cultured in DMEM supplemented with 10% FBS, 1% penicillin, and streptomycin.

### 2.10. Organoid Viability Assay

Organoids were dissociated into single cells, and 600 viable cells were seeded per well in 50 μL (50% Matrigel/50% human complete organoid media). KV and/or oxaliplatin were added 72 h after seeding. After 3 days, cell viability was assessed by 3D Cell Titer-Glo^®^ according to the manufacturer’s instructions (Promega, Madison, WI, USA) on a Luminoskan Ascent (Thermo Fisher) plate reader. The Cell Titer-Glo^®^ 3D Cell Viability Assay is a homogeneous method to determine the number of viable cells in 3D cell culture based on quantitation of the ATP present, which is a marker for the presence of metabolically active cells. A 50% inhibitory concentration (IC_50_) was defined as the drug concentration that inhibited cell growth by 50% compared to the untreated control. Dose–response curves were generated by luminescence. The area under the curve (AUC) was calculated and normalized by dividing the AUC value by the maximum area.

### 2.11. Mutation Profile of PDPCOs

Paired-end sequences were first mapped to the human genome by a HiSeq Instrument, where the reference sequence was UCSC assembly hg19 (original GRCh37 from NCBI, 27 February 2009), using the mapping program BWA (version 0.7.12), and a mapping result file was generated in BAM format using BWA-MEM. Then, Picard-tools (ver.1.130) were applied to remove polymerase chain reaction duplicates. The local realignment process was performed to locally realign reads with BAM files. By using the Genome Analysis Tool kit, base quality score recalibration and local realignment around indels were performed. Haplotype Caller of GATK was used for variant genotyping for each sample based on the BAM file previously generated (SNP and short indels candidates were detected). These variants were annotated by SnpEff v4.1g, to vcf file format, filtering with dbSNP for the version of 142. Then, SnpEff was applied to filter additional databases, including ESP6500, ClinVar, and dbNSFP 2.9.

### 2.12. Data Analysis

IC_50_ value was then estimated using the fitted line: Y = a X + b, where Y = 0.5b, IC_50_ = X = −0.5b/a. All data represent the mean ± SD from at least three independent experiments. A student’s *t*-test was used to identify significant differences between the two groups.

## 3. Results

### 3.1. Krukovine Shows an Antiproliferative Effect toward KRAS-Mutated Pancreatic Cancer Cells and Oxaliplatin-Resistant Pancreatic Cancer Cells

*KRAS* is the most frequent genetic alteration in pancreatic cancer (>90%). AsPC-1, Panc-1, and MiaPaCa-2 cell lines with the *KRAS* mutation were treated with KV at 0, 12.5, 25, and 50 μM for 48 h. Results showed that krukovine (KV) (Figure 1A) inhibited the growth of all tested pancreatic cancer cell lines in a dose-dependent manner (Figure 1B). Meanwhile, BxPC-3 (non-*KRAS* mutated pancreatic cancer cells) showed resistance to KV treatment compared to *KRAS*-mutated pancreatic cancer cells (Figure 1B).

To further evaluate the effects of KV on oxaliplatin-resistant pancreatic cancer cells, AsPC-1 cells were tested by increasing the concentrations of oxaliplatin treatment in the AsPC-1 cells. The AsPC-1 cells (IC_50_ > 100 µM) that exhibited higher resistance to oxaliplatin (Figure 1C) were selected. However, KV exerted potent antiproliferative activity against oxaliplatin-resistant AsPC-1 cells with similar IC_50_ values to the original AsPC-1 cells and the other pancreatic cancer cells (Figure 1D). These data suggest that KV may potentially overcome oxaliplatin resistance in pancreatic cancer cells.

### 3.2. RNA Levels of KV-Treated Pancreatic Cancer Cells Show Major Metabolic Pathway

To explore the anticancer properties of KV on oxaliplatin-resistant cancer cells, we examined the RNA level of KV-treated oxaliplatin-resistant AsPC-1 cells versus KV-untreated controls. These results are shown in Appendix A. Appendix A shows the major pathway in the KV-treated sample and shows top KEGG pathways enriched in oxaliplatin-resistant AsPC-1 cells such as PI3K-Akt signaling, MAPK, multicellular organism, and metabolism in the cancer pathway (*p* value < 0.05 (*), 0.01 (**), 0.001 (***)). Furthermore, the RNA-seg result showed that the TMEM139 expression level was highly downregulated in KV-treated oxaliplatin-resistant AsPC-1 cells versus KV-untreated controls (Table 1).

### 3.3. The Clinical Significance of TMEM139 Expression in Pancreatic Cancer Patients

A growing body of evidence suggests that transmembrane139 (TMEM139) expression is highly associated with cancer development and metastasis in a variety of cancers [10,27]. Furthermore, TMEM139 gene was identified as the risk prediction marker of pancreatic adenocarcinoma patients by energy metabolism characteristics [29]. To confirm the involvement of TMEM139 expression levels in human pancreatic cancers, the clinical significance of TMEM139 expression in pancreatic cancer patients was analyzed using the Kaplan−Meier method for overall survival (OS) and relapse-free survival (RFS). As shown in Figure 2, higher levels of TMEM139 expression were associated with a decreased probability for OS and RFS compared with lower levels of TMEM139 expression in patients with pancreatic cancer. These data indicate that the expression levels of TMEM139 are negatively correlated with OS and RFS in patients with pancreatic cancer. These findings suggest that TMEM139 might be a therapeutic target for the management of pancreatic cancer.

### 3.4. Effects of KV on the TMEM 139 Expression and TMEM139-Associated Signaling Pathway in Oxaliplatin-Resistant AsPC-1 Cells

To confirm whether the treatment of KV in oxaliplatin-resistant AsPC-1 cells is associated with the expression of TMEM139 biomolecules, oxaliplatin-resistant AsPC-1 cells were treated with KV for 48 h, and the corresponding mRNA and protein expressions were evaluated by RNA sequencing (Appendix A) and Western blot analysis, respectively. Western blot results were from duplicate or triplicate samples per condition. KV significantly suppressed the mRNA and protein expression of TMEM139 in oxaliplatin-resistant AsPC-1 cells (Table 1; mRNA fold change, Figure 3A; Western blot analysis).

The protein–protein interaction online database search tool (STRING) showed that TMEM139 is related to the RPS6K and MAPK3/1 (Erk1/2) signaling pathways (Figure 3B; protein–protein interaction). The downregulation of mRNA expression of TMEM139 by KV treatment was found in oxaliplatin-resistant AsPC-1 cells (Table 1). The protein expressions of p-ERK1, 2 (Thr202/Thy204), RPS6K, and TMEM139 were downregulated by KV in oxaliplatin-resistant AsPC-1 cells (Figure 3A). These data suggest that the antiproliferative activity of KV may be in part associated with the suppression of the MAPK (Erk1/2) pathway with the Erk-RPS6K-TMEM139 axis in oxaliplatin-resistant AsPC-1 cells (Figure 3A).

### 3.5. KV Inhibits the PI3K-Akt-mTOR Pathway in Oxaliplatin-Resistant AsPC-1 Cells

We also evaluated the effect of KV on mediating the PI3K-Akt-mTOR (Figure 3C; protein–protein interaction; PI3K-Akt and PI3K-Akt-mTOR) pathway. The mRNA expression of the MAPK and PI3K-Akt signaling pathways were differentially expressed in KV-treated oxaliplatin-resistant AsPC-1 cells (Appendix A). The expression levels of phospho-PI3K, phospho-AKT (Ser473), phospho-mTOR, total-PI3K, total-AKT, and total-mTOR were determined by Western blotting (Figure 3A). Results indicated that KV decreased the levels of phospho-PI3K, phospho-AKT, and phospho-mTOR, but did not significantly affect the levels of total-PI3K, total-AKT, and total-mTOR (Figure 3A). The uncropped blots are shown in Appendix A.

### 3.6. KV Inhibits Migration and Invasion in Oxaliplatin-Resistant AsPC-1 Cells

The oxaliplatin-resistant AsPC-1 cells treated with KV were evaluated via RNA-seq, scratch wound healing assay, and transwell analysis. KV significantly inhibited wound healing ability (Figure 4A) and transwell invasion ability (Figure 4B) in the oxaliplatin-resistant AsPC-1 cells in a dose-dependent manner. The KV-treated RNA sequencing data showed that the data were related to cell migration and invasion processes. (Figure 4C).

### 3.7. KV Showed an Antiproliferative Effect and Enhanced the Anticancer Effects of Oxaliplatin in KRAS-Mutated Patient-Derived Pancreatic Cancer Organoids

All organoids underwent NGS (Next-Generation Sequencing) and had the *KRAS* mutation (Figure 5A). The multidrug-resistant or -sensitive organoids (relatively resistant; SNU-4425-TO, SNU-4340-TO, and SNU-3947-TO, relatively sensitive; SNU-4305-TO and SNU-4206-TO) were selected (Appendix A) and treated with KV at 0, 6.25, 12.5, 25, and 50 μM for 72 h (Figure 5B). Normal mouse organoid (non-cancer organoid) showed to be relatively sensitive to KV treatment. Results showed that KV inhibited the growth of all tested PDPCOs in a dose-dependent manner (Figure 5B,C). Oxaliplatin combined with KV was found to be more effective in inhibiting *KRAS*-mutated organoid growth compared to the individual drugs (Figure 5D,E). The combinatorial effect of KV and oxaliplatin in PDPCO was identified by the CI (combination index). SNU-4425-TO showed to be more oxaliplatin-resistant than KV; however, the combinations of oxaliplatin and KV exhibited synergy (CI range, 0.3–0.7, synergism) (combination index: Antagonism (CI > 1), additivity (CI = 1) and synergism (CI < 1)). Furthermore, the expression levels of apoptosis-related proteins were measured by Western blot to explore the anticancer properties of oxaliplatin combined with KV. The results showed that oxaliplatin combined with KV increased the expression of cleaved PARP (poly ADP ribose polymerase) in PDPCO (SNU-4435-TO) (Figure 5F). The uncropped blots are shown in Appendix A.

## 4. Discussion

Oxaliplatin is commonly used as the first-line chemotherapeutic agent in the FOLFIRINOX (leucovorin + 5-fluorouracil + oxaliplatin + irinotecan) regimen for treating pancreatic cancer in the clinical setting. Therefore, oxaliplatin resistance in pancreatic cancer patients is highly associated with failures in chemotherapy. The search for novel agents that can offset oxaliplatin resistance while resulting in minimal side effects thus represents a major goal in pancreatic cancer research. For many years, natural products have been considered as potential resources for novel and safe drug discovery [18]. In this study, we used oxaliplatin-resistant pancreatic cancer cells (oxaliplatin-resistant AsPC-1 cells) and showed the inhibitory effect of the natural compound, KV, on the oxaliplatin-resistant pancreatic cancer cells.

Most pancreatic cancer patients receive diagnosis at stages 3 and 4, both of which are advanced metastatic states [3]. However, currently available therapeutic options in the management of metastatic pancreatic cancer are highly limited, leading to drug response failure [30]. Therefore, there is an urgent need for effective anti-metastatic therapeutic agents to treat pancreatic cancer patients. Extracts from natural products such as Toosendanin and Pao Pereira are known for the inhibition of pancreatic cancer metastasis and invasion [31,32]. In this study, KV isolated from the bark of *Abuta grandifolia* (Mart.) Sandw. (Menispermaceae) inhibited TMEM139 expression (mRNA and protein), which is associated with cancer metastasis, and with the migration and invasion abilities in oxaliplatin-resistant AsPC-1 cells. The clinical significance of TMEM139 expression in pancreatic cancer patients was analyzed using the Kaplan−Meier method for overall survival (OS) and relapse-free survival (RFS). The expression levels of TMEM139 are negatively correlated with OS and RFS in patients with pancreatic cancer. These results showed that KV can be an effective antimetastatic reagent by inhibiting TMEM139 expression for pancreatic cancer patients. However, bias in the Kaplan–Meier survival estimation can arise from various sources, including incomplete follow-up, differential censoring, and informative censoring. Therefore, to minimize bias, further study is required. Furthermore, the wound healing and invasion assay have limitations, such as difficulty in controlling variables and the potential for artifacts. Therefore, in-depth study will be needed to validate those results.

Genetic alterations can be found in approximately 97% of pancreatic ductal adenocarcinoma cases [13]. Whole-genome sequencing has shown the main driver genes in pancreatic cancers such as *KRAS*, *CDKN2A*, *TP53*, and *SMAD* [33]. These genes are mutated in different stages and their dysregulation promotes the differentiation and proliferation of pancreatic cancers [33]. *KRAS* mutations generally emerge from stage 1 lesions (PanIN-1) to promote the PDAC initiation process [13,33]. Mutations in the *KRAS* gene regulate cell proliferation, differentiation, and apoptosis via activation of downstream signal transduction pathways [13]. The early-phase drug trials in pancreatic cancer have therefore largely focused on the indirect inhibition of *RAS*, in particular, targeting downstream signaling such as the RAF-MEK-ERK pathway [13]. The RAF-ERK and PI3K-AKT pathways are two major hyper-activated downstream pathways in *KRAS* mutation, which induce the uncontrolled proliferation of cancer cells and metastasis in cancer patients [34,35]. Therefore, these signaling pathways have been identified as promising targets in cancer therapy [36]. KV is a pure compound from a plant extract that exhibits anticancer/antitumor effects by inactivating AKT signaling and inhibiting the RAF-ERK signaling pathway in *KRAS*-mutated lung cancer cells [12]. In this study, we discovered the effect of KV on mediating the PI3K-Akt-mTOR and Erk-RPS6K-TMEM139 signaling pathways in oxaliplatin-resistant AsPC-1 cells (AsPC-1 cells; *KRAS*-mutated pancreatic cancer cells).

Immunotherapy has recently shown great potential to transform future PDAC treatment. However, PDAC has shown inferior treatment outcomes toward various immunotherapy regimens compared to other cancer types [37]. The reason is the tumor microenvironment (TME), which has been considered as the fundamental underlying barrier to treatment resistance [37]. The heterogeneity of PDAC composition has led to failures in clinical trials. Current in vitro techniques cannot sufficiently replicate the pancreatic ductal adenocarcinoma tumor microenvironments. The development of patient-derived organoids has opened the door for tumor microenvironment study since PDOs are derived from patient tumors, thus preserving the tumors’ unique genetic aspects and phenotype; furthermore, they can be cocultured with immune cells, fibroblasts, and other stromal cells [38]. In this study, patient-derived pancreatic cancer organoids (PDPCOs) were used to evaluate KV anticancer activity. We explored the *KRAS* mutation of PDPCOs and tested multi-anticancer drugs on PDPCOs to select the *KRAS*-mutated multidrug-resistant organoids. The results confirmed the anticancer potential of KV in PDPCOs.

Nowadays, a combination of drugs is increasingly favored over single agents in the treatment of pancreatic cancer patients [39,40,41]. In this study, we evaluated the combinatorial effect of KV with oxaliplatin and found a synergistic effect. The results showed that KV used in combination with oxaliplatin increases the cleaved-PARP expression in PDPCOs, indicating that KV combined with oxaliplatin contributes to the inhibition of PDCPO proliferation. Based on the combinatorial effect of KV with oxaliplatin, this compound may be an effective combinatorial therapeutic agent for the treatment of *KRAS*-mutated pancreatic cancer. Therefore, one of the potential mechanisms of the combination effect of KV and oxaliplatin in inhibiting PDAC could be inhibiting cancer stem cells by upregulating cleaved-PARP expression as in the case of PDPCO treated with KV and oxaliplatin in this study.

In this study, instead of using an animal study, we used pancreatic cancer patient-derived organoids to validate the KV efficacy. The AsPC-1 cells are relatively sensitive to KV and the most resistant to oxaliplatin among the tested pancreatic cancer cell lines. Therefore, we selected AsPC-1 cells for further study. However, ASPC-1 cells are derived from PDAC patient ascites which have mesenchymal features, and thus are not proper for epithelial–mesenchymal transition (EMT) study. Therefore, other pancreatic cancer cell lines are needed for EMT study.

Taken together, it can be concluded that KV has great potential as a novel therapeutic agent for future use in *KRAS*-mutated and oxaliplatin-resistant pancreatic cancer patients, based on its inhibitory activity of the Erk-RPS6K-TMEM139 signaling pathway in *KRAS*-mutated and oxaliplatin-resistant pancreatic cancer cells. Based on the combinatorial effect of KV with oxaliplatin, this compound may be an effective combinatorial therapeutic agent for the treatment of *KRAS*-mutated pancreatic cancer.

## 5. Conclusions

In our study, we investigated the therapeutic effects of KV to inhibit the growth and proliferation of *KRAS*-mutated pancreatic cancer cell lines (AsPC-1, Panc-1, and MiaPaCa-2) and oxaliplatin-resistant pancreatic cancer cells (oxaliplatin-resistant AsPC-1 cells). KV effectively inhibited TMEM139 expression and proliferation via the Erk-RPS6K-TMEM139 pathways in oxaliplatin-resistant AsPC-1 cells. KV also inhibited the invasion and migration of oxaliplatin-resistant AsPC-1 cells. Furthermore, KV showed a growth inhibitory effect on *KRAS*-mutated PDPCOs and a combinatory effect with oxaliplatin in PDPCO.

## Figures and Tables

**Figure 1 cancers-15-02642-f001:**
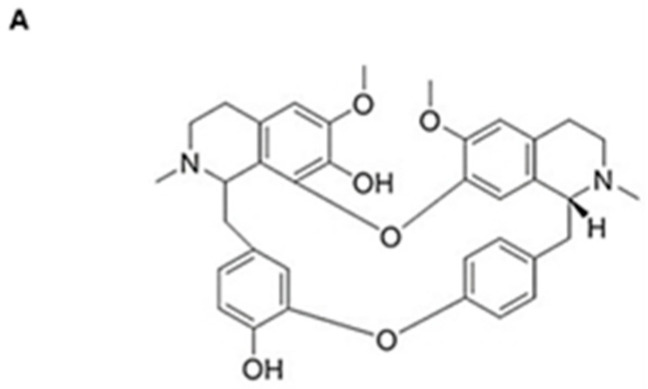
(**A**) The structure of krukovine. (**B**) Cell viability of KV-treated pancreatic cancer cell lines. The viability of pancreatic cancer cells was measured using the MTT assay following treatment with the indicated concentrations of KV for 48 h. All data are presented as mean ± SD (*n* = 3, * *p* < 0.05, *** *p* < 0.001). (**C**) Oxaliplatin-resistant and non-oxaliplatin-resistant AsPC-1 cells; microscopic images. Scale bar: 50 μm. Microscopic images of AsPC-1 cells were measured using the light microscope following treatment with the indicated concentration of oxaliplatin (100 μM) for 48 h. (**D**) Cell viability of oxaliplatin-resistant AsPC-1 cells treated with oxaliplatin and KV. The viability of oxaliplatin-resistant AsPC-1 cells was measured using the MTT assay following treatment with the indicated concentrations of oxaliplatin or KV for 48 h. Data are presented as the mean ± SD (*n* = 3, * *p* < 0.05, *** *p* < 0.001).

**Figure 2 cancers-15-02642-f002:**
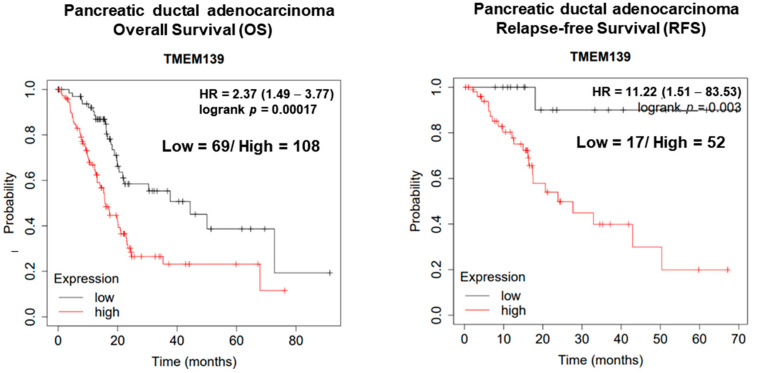
The Kaplan–Meier survival curve according to the TMEM139 expression level. The Kaplan–Meier survival curve represents the overall survival (OS) and relapse-free survival (RFS) of pancreatic cancer patients according to the TMEM139 expression level.

**Figure 3 cancers-15-02642-f003:**
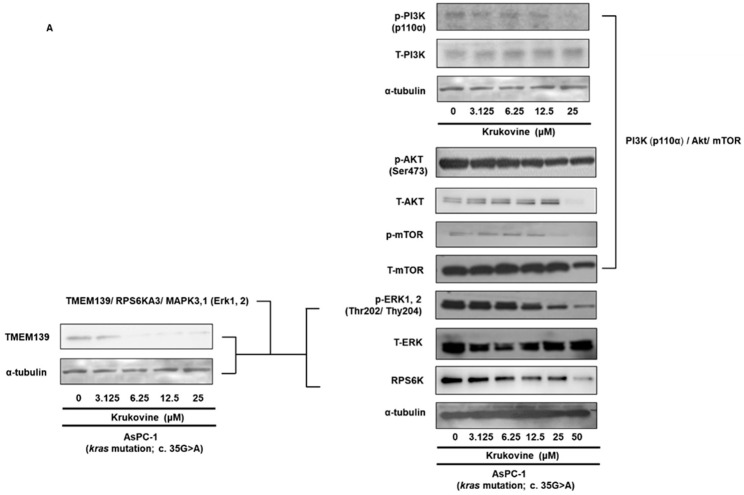
(**A**) The effect of KV on TMEM139 protein, the TMEM139-associated signaling pathway, and the PI3K-Akt-mTOR signaling pathway in oxaliplatin-resistant AsPC-1 cells. Oxaliplatin-resistant AsPC-1 cells were treated with the indicated concentrations of KV for 48 h, and the protein expression level of TMEM139 protein, the TMEM139-associated signaling pathway, and the PI3K-Akt-mTOR signaling pathway was determined by Western blotting. α-tubulin was used as an internal control. (**B**) Protein–protein interaction (PPI) for TMEM139 from the search tool (STRING) database. Interacting proteins for *TMEM139* (*Transmembrane 139*) Gene: MAPK3/1 (Mitogen-Activated Protein Kinase 3/1; Erk1/2)-RPS6KA3 (Ribosomal Protein S6 Kinase A3)-TMEM139 (**C**) Protein–protein interaction (PPI) for PI3K-Akt and PI3K-Akt-mTOR signaling pathway from the search tool (STRING) database. Interacting proteins for *Akt1* (*AKT Serine/Threonine Kinase*) gene: PI3K-Akt and PI3K-Akt-mTOR.

**Figure 4 cancers-15-02642-f004:**
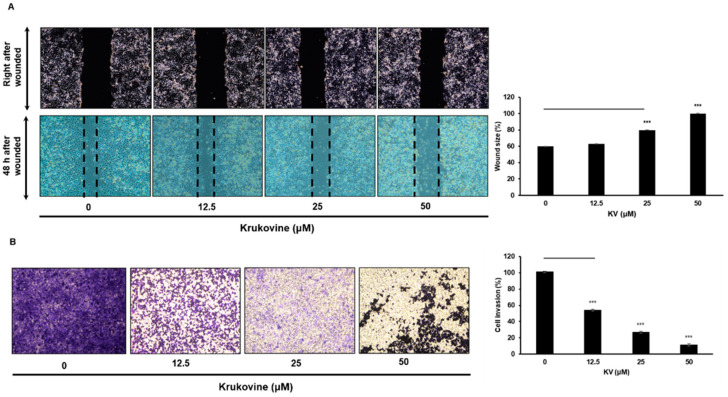
(**A**) The effect of KV on oxaliplatin-resistant AsPC-1 cells’ migration. Monolayers of oxaliplatin-resistant AsPC-1 cells were scratched mechanically and treated with KV (indicated concentrations) for 48 h. Representative images of wound closure obtained under a light microscope. All data are presented as mean ± SD (*n* = 3, *** *p* < 0.001). (**B**) The effect of KV on oxaliplatin-resistant AsPC-1 cell invasion. Oxaliplatin-resistant AsPC-1 cells were pretreated with indicated concentrations of KV for 24 h, reseeded into the upper chamber of transwell inserts, and incubated for 24 h. The cells that invaded lower chambers were then fixed, stained, imaged, and counted. All data are presented as mean ± SD (*n* = 3, *** *p* < 0.001). (**C**) Top 20 terms of GO (Gene Ontology) functional analysis. RNA-seq assay was performed in oxaliplatin-resistant AsPC-1 cells treated with KV (25 μM) and compared with a control (PBS with DMSO-treated). The dot color represents the *p*-values. The scale of the spots indicates the number of genes involved.

**Figure 5 cancers-15-02642-f005:**
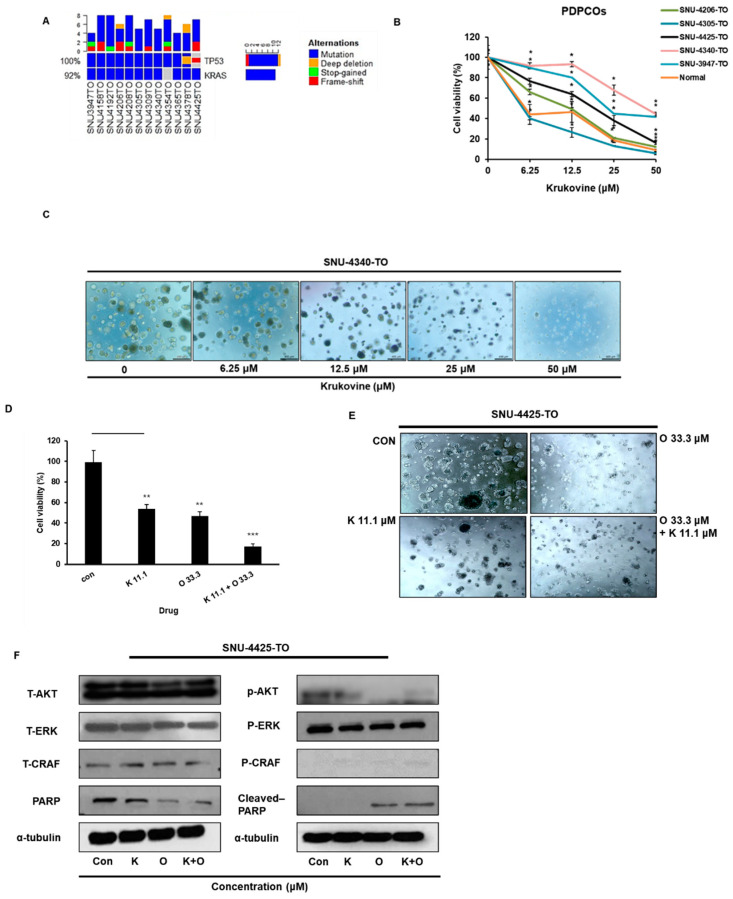
(**A**) Mutation profile of PDPCOs. The main representative mutations (*KRAS* and *TP53*) in PDPCOs (patient-derived pancreatic cancer organoids) were analyzed through whole-exome sequencing analysis. (**B**) Antiproliferative effect of KV on PDPCOs. PDPCOs (SNU-4206, SNU-4305-TO, SNU-4425-TO, SNU-3947-TO, and SNU-4340-TO) were treated with various concentrations of KV and detected by 3D cell titer glow assay after 72 h. All data are presented as mean ± SD (*n* = 3, * *p* < 0.05, ** *p* < 0.01, *** *p* < 0.001). (**C**) KV inhibited the growth of the tested PDPCO (SNU-4340-TO) in a dose-dependent manner. Representative images of the organoid treated with KV obtained under a light microscope. (**D**) Cell viability assay for measuring the combination effect of KV with oxaliplatin. The multidrug-resistant organoid (SNU-4425-TO) was treated with KV and/or oxaliplatin for 72 h with indicated concentration (con; control, K 11.1; krukovine 11.1 μM, O 33.3; oxaliplatin 33.3 μM, K 11.1 + O 33.3; krukovine 11.1 μM + oxaliplatin 33.3 μM) and detected by 3D cell titer glow assay. All data are presented as mean ± SD (*n* = 3). All data are presented as mean ± SD (*n* = 3, * *p* < 0.05, ** *p* < 0.01, *** *p* < 0.001). (**E**) Combination effect of KV with oxaliplatin on the growth of multidrug-resistant PDPCO (SNU-4425-TO). Representative images of the organoid treated with KV and/ or oxaliplatin for 72 h with the indicated concentrations (CON; control, K 11.1; krukovine 11.1 μM, O 33.3; oxaliplatin 33.3 μM, K 11.1 + O 33.3; krukovine 11.1 μM + oxaliplatin 33.3 μM). Images were obtained under a light microscope. (**F**) Effects of KV on the cleaved-PARP expression in multidrug-resistant organoid (SNU-4425-TO). SNU-4425-TO were treated with the indicated concentrations of KV for 72 h, and the protein expression level was determined by Western blotting. α-tubulin was used as an internal control.

**Table 1 cancers-15-02642-t001:** TMEM gene family mRNA expression.

Gene Symbol	Description	kruko/con.fc	kruko/con.raw.pval
** *TMEM139* **	**transmembrane protein 139**	**−20.08857481**	**0.000184391**
*TMEM160*	transmembrane protein 160	−10.07644718	0.0002668
*TMEM102*	transmembrane protein 102	−9.08294669	0.000505557
*TMEM187*	transmembrane protein 187	−7.717754905	0.001085602
*TMEM221*	transmembrane protein 221	−7.22073532	0.019434951
*TMEM249*	transmembrane protein 249	−6.628590914	0.006072763
*TMEM177*	transmembrane protein 177	−5.850793145	0.004004917
*TMEM92*	transmembrane protein 92	−5.793088235	0.005077808
*TMEM121*	transmembrane protein 121	−5.624075595	0.005226063
*TMEM203*	transmembrane protein 203	−4.929095646	0.008670705
*TMEM223*	transmembrane protein 223	−4.591863583	0.011703338
*TMEM238*	transmembrane protein 238	−4.352286379	0.028704807
*TMEM191B*	transmembrane protein 191B	−3.783402356	0.034476139
*TMEM191A*	transmembrane protein 191A	−3.7829524	0.031869069
*TMEM53*	transmembrane protein 53	−3.782392019	0.027051
*TMEM191C*	transmembrane protein 191C	−3.556386263	0.043266828
*TMEM141*	transmembrane protein 141	−3.384076799	0.040373276
*TMEM115*	transmembrane protein 115	−3.287999353	0.045333404
*TMEM186*	transmembrane protein 186	−3.242921449	0.049370419

TMEM family mRNA expression (fold change) and *p*-value in oxaliplatin-resistant AsPC-1 cells treated with KV.

## Data Availability

Data is contained within the article. The data presented in this study are available in Section 2.

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
