# Peer review of "Antiproliferative Activity of Krukovine by Regulating Transmembrane Protein 139 (TMEM139) in Oxaliplatin-Resistant Pancreatic Cancer Cells"

_cancers, 2023, doi:10.3390/cancers15092642_

Round 1

Reviewer 1 Report

This study investigated the anticancer efficacy and mechanism of Krukovine (KV), an alkaloid isolated from the bark of Abuta grandifolia, in oxaliplatin-resistant pancreatic cancer cells and patient-derived pancreatic cancer organoids (PDPCOs) with Kras mutation. KV was found to suppress tumor progression through downregulation of the Erk-RPS6K-TMEM139, PI3K-Akt-mTOR pathways, and EMT signals. Additionally, KV showed an antiproliferative effect in PDPCOs, and the combination of KV and oxaliplatin inhibited PDPCO growth more effectively than either drug alone.

Major:

  1. Regarding the Western blot: It is essential to include the whole Western blot in the Supplementary Materials section for several reasons. First, it allows readers to verify the authenticity of the presented data and ensure that the results are not misinterpreted or selectively reported. Second, it enables other researchers to compare the results of the current study with their own data, which could lead to new insights and collaborations. Third, it serves as a reference for future studies that may build upon the current findings. Proper methodology for Western blotting includes using appropriate positive and negative controls, ensuring adequate sample preparation, selecting an appropriate loading control, and using standardized protocols for blotting and detection. Limitations of Western blotting include variability due to sample preparation, potential for nonspecific binding, and difficulties in quantification.

  2. Wound healing (scratch assay) and invasion migration assay: The scratch assay is a widely used in vitro method to assess cell migration and wound healing. It involves creating a scratch in a cell monolayer and monitoring the migration of cells into the wound area over time. Limitations of the scratch assay include variability in scratch creation, variability in cell seeding density, and potential for interference from cell proliferation. To minimize these limitations, it is important to standardize the protocol for scratch creation, use consistent cell seeding density, and control for cell proliferation by monitoring the cells over a specific time frame. Invasion and migration assays are used to assess the ability of cancer cells to invade and migrate through extracellular matrix. These assays can be performed in vitro using Transwell chambers or Boyden chambers. Proper methodology includes using appropriate controls, standardizing the cell seeding density, and quantifying the number of cells that have invaded or migrated.

  3. Colony forming unit assay normalization criteria: The colony forming unit (CFU) assay is a widely used method to assess the clonogenic potential of cancer cells. It involves plating cells in low-density conditions and allowing them to grow into colonies over time. Proper normalization criteria include using an appropriate cell density for plating, selecting a time point for counting the colonies, and using a consistent staining and counting method. It is also important to use appropriate statistical methods to analyze the data.

  4. Bias analysis of Kaplan-Meier survival estimation: Kaplan-Meier survival estimation is a commonly used method to analyze survival data in cancer research. Bias can arise from various sources, including incomplete follow-up, differential censoring, and informative censoring. To minimize bias, it is important to use appropriate statistical methods, such as inverse probability of censoring weighting, and to report the assumptions made in the analysis. Sensitivity analyses can also be used to assess the robustness of the results to different assumptions.

  5. The Cox proportional hazards model is a statistical method commonly used to analyze time-to-event data, such as survival data. It allows for the estimation of the hazard ratio, which is a measure of the relative risk of an event occurring in one group compared to another group, while controlling for other variables.

    In the context of cancer research, the Cox model can be used to analyze the association between various factors and patient survival. The model takes into account multiple variables, such as age, sex, tumor stage, and treatment, and can provide adjusted hazard ratios that reflect the impact of each variable on patient survival while controlling for the effects of other variables.

    To ensure the validity of the results obtained from the Cox model, it is important to carefully select the variables included in the model and to assess the proportional hazards assumption, which states that the hazard ratio should remain constant over time.

    Furthermore, it is essential to validate the results obtained from the Cox model using independent datasets and to perform sensitivity analyses to test the robustness of the findings. Biases in data collection, measurement, and analysis should also be carefully considered and addressed.

    Overall, the Cox proportional hazards model is a powerful tool for analyzing survival data in cancer research, but it should be used with caution and in conjunction with other statistical methods to ensure the validity and reliability of the results.

  6. The reviewer has noted that there is important information missing in the introduction and discussion sections of the paper. Specifically, the paper lacks a discussion on genetic alterations, especially the K-Ras mutation, which is known to play a significant role in the progression of pancreatic precursor lesions into pancreatic ductal adenocarcinoma (PDAC). Additionally, the tumor microenvironment is one of the major challenges that hinder therapeutic approaches from functioning sufficiently and leads to immune evasion of pancreatic malignant cells. Understanding the mechanisms of these two hallmarks of PDAC is crucial for developing effective treatment strategies.

    To further elaborate, the genetic alterations and mutations that occur in PDAC are among the most well-known and researched aspects of this disease. Specifically, the K-Ras mutation is present in the majority of PDAC cases and plays a key role in the initiation and progression of the disease. Therefore, it is important to discuss the implications of K-Ras mutations and their impact on PDAC pathogenesis and treatment.

    Additionally, the role of the tumor microenvironment in PDAC is an emerging area of research that is gaining increasing attention. The tumor microenvironment, which includes immune cells, fibroblasts, and other stromal cells, plays a significant role in promoting tumor growth and resistance to therapy. Therefore, it is important to discuss the role of the tumor microenvironment in PDAC and its implications for therapeutic approaches.

    Finally, the paper does not discuss the potential limitations and biases of the research methods used in the study, which could affect the validity and generalizability of the results. For example, the scratch assay and invasion migration assay have limitations, such as difficulty in controlling variables and the potential for artifacts. Similarly, the normalization criteria used for the colony forming unit assay should be carefully considered to ensure accurate interpretation of the results. Additionally, the Kaplan-Meier survival analysis should be complemented with multivariable Cox regression analysis to adjust for potential confounders and assess the independent effect of the treatment on survival.

    Overall, the paper would benefit from a more comprehensive and thorough discussion of these important topics, which could enhance the reader's understanding of the study's findings and their implications for PDAC treatment. Refer to

  7. PMID: 33918146 and expand
  8. Suggestions regarding point 6: In this study, the authors investigated the antiproliferative activity of Krukovine (KV) in oxaliplatin-resistant pancreatic cancer cells and explored the mechanism of action. They found that KV suppresses tumor progression via the downregulation of the Erk-RPS6K-TMEM139 signaling pathway.

    Transmembrane protein 139 (TMEM139) has been identified as a novel oncogene and its overexpression has been associated with various types of cancer, including pancreatic cancer. In this study, the authors found that KV treatment significantly downregulated TMEM139 expression in oxaliplatin-resistant pancreatic cancer cells.

    The authors suggest that KV could be a potential therapeutic agent for the treatment of pancreatic cancer, particularly for patients with Kras mutations and oxaliplatin-resistant tumors. They propose that the downregulation of TMEM139 by KV may contribute to its antiproliferative activity and could be a potential target for future therapeutic interventions.

    Overall, this study provides insight into the mechanism of action of KV and its potential as a therapeutic agent for pancreatic cancer treatment, particularly for patients with oxaliplatin-resistant tumors. Further studies are needed to validate these findings and to investigate the clinical efficacy of KV in pancreatic cancer patients.

Reviewer 2 Report

A brief summary:

The manuscript is clear and could educate the readers with a new avenue of treatment.

The research article attempts to demonstrate the significance of Krukovine (KV) in oxaliplatin-resistant pancreatic cancer with Kras mutation. KV is shown by the authors to have the capability to significantly kill oxaliplatin-resistant pancreatic cancer cells. The implication of the treatment to downregulate key pathways involving Erk, mTOR and TMEM139 is quite interesting, especially given the implication of TMEM139 expression on the clinical outcome of the patients. Ability of KV to reduce the viability of patient-derived pancreatic cancer organoids lends this article high clinical relevance.

General concept comments

Overall, this article has great potential and value to the readers of this journal. However, author needs to address key concerns as listed below.

1) On page 1, under Introduction, the author could consider citing available statistics to demonstrate what percentage of the patients develop acquired resistance to oxaliplatin. That would make the significance of KV more evident.  

2) On page 8 and 9, under results (3.4) for the western blots that have been reported, authors have not mentioned if these have been reproduced and replicated. The authors should consider describing how many biological replicates were involved and how many times were the western blots reproduced. The readers could benefit from that information as well as from quantification of the western blots.

3) On page 10, under figure 4, it seems the figure 4(A) and 4(B) are missing the graphs that were associated with them and reported in the figure legends. Only the axis were visible not the data. Hence, it could not be reviewed. While for Figure 4(D) the same comments about western blot replication and quantification apply as that for point 2.

4) The text in all the figures needs to be bigger and legible. Especially figure 4 that is missing data. The legends were barely readable in the printed version of the article.

Reviewer 3 Report

In the manuscript” Antiproliferative Activity of Krukovine by Regulating of Transmembrane protein 139 (TMEM139) in Oxaliplatin-resistant Pancreatic Cancer Cells”, the authors concluded that KV suppresses tumor progression via the downregulation of the Erk-RPS6KTMEM139, PI3K-Akt-mTOR pathways, and EMT signals. Furthermore, KV showed the antiproliferative effect in PDPCOs and the combination of OXA and KV inhibited PDPCO growth more effectively than either drug alone.

In general, the study is well designed. The language is overall clear, however, there are several redundancies, grammatical errors, and typos.  The methodology is overall solid. The figures are overall well-made and labeled. The underlining mechanism study is not sufficient. The authors should have an in-depth discussion on the limitations and some inconsistencies in the study.

Overall, the data can partially support the conclusion. In my opinion, the current evidence did not support the inhibition of EMT by KV.

Here are the major points the author needs to revise or clarify:

1.     In addition, the author should consider adding the normal pancreatic ductal cells lines, HPNE, for the treatment and comparison.

2.     For figure 1B, the images were described as figure 1C in the figure legends. In addition, please provide higher resolution images.

3.     For figure 1B, how did the author define or generate oxaliplatin-resistant AsPC-1 cells? In the images, there is a “ASPC-1” phenotype and an “oxaliplatin-resistant AsPC-1” phenotype. Did the authors screen an oxaliplatin-resistant AsPC-1 cell phenotype? Please describe it in detail. It is confusing here.

4.     For figure 1, what is the rational of using ASPC-1 cell lines? ASPC-1 cells are derived from PDAC patient ascites. It has higher mesenchymal features and does not mimic the overall features, including EMTs, of PDAC cells in situ. The authors need to have an in-depth discussion.

5.     In 3.2, did the author find TMEM139 expression changed in KV treatment aspc-1 cells? There is no logical connection from 3.2 to 3.3. The author may delete 3.2 or have some in depth analysis from the KEGG analysis.

6.     For figure 4A B, the quantitative bar graph is missing.

7.     For figure 4A B, the authors previously showed that KV significantly inhibited the proliferation of oxaliplatin-resistant AsPC-1 cells at 25 and 50uM. The authors need to normalize the migration/invasion according to the proliferation inhibition.

8.     In figure 4D, the authors need to provide better images for WB. The expression of Snail seems to have increased with KV 50uM. The expression of Twist seems to gradually increase with treatment. The authors need to explain the inconsistency. In addition, the authors need to show the WB data on E-cad.

9.     In figure 4D, with only N-cad decreased, I suggested the authors do not claim EMT inhibition. As mentioned in 4, ASPC-1 cells are derived from ascites, with already have higher mesenchymal features and usually don’t undergo EMT any further. If the author concludes the EMT was inhibited, an epithelial phenotype cells lines is needed as a comparison.

10.  In figure 5, the authors have a combined treatment of KV and Oxaliplatin. The authors should have combined treatment for other parts of the study (e.g., proliferation, migration, invasion, WT on EMT, PI3K pathways). As the author investing oxaliplatin-resistance, a combination treatment is needed.

Minor:

1.     For figure 1B, please use “proliferation” instead of “absorbance” in MTT assay.

Round 2

Reviewer 1 Report

This reviewer has still the feeling that a larger part should be dedicated to introduce and discus how pancreatic ductal adenocarcinoma (PDAC) is primarily driven by genetic alterations, particularly the K-Ras mutation, which poses a significant obstacle to its treatment. However, the tumor microenvironment also plays a crucial role in hindering therapeutic approaches and enabling immune evasion of malignant pancreatic cells. Understanding the mechanisms underlying both of these hallmarks of PDAC is essential in overcoming the challenges in its treatment: the signaling pathways implicated in PDAC development and the immune system's involvement in pancreatic cancer, including the potential of immune checkpoint inhibition as a promising next-generation therapeutic strategy. Targeting the signaling and immune checkpoint molecules involved, alone or in combination with conventional therapies, has yielded the most promising results in treating pancreatic cancer. The last decision is for the editor.

Reviewer 2 Report

For the western blots you responded that you did not do many replications due to the limited amount of compound (Krukovine).

As the representative images you have shown only have one sample per condition. describing if these results are from single sample per condition, duplicate triplicate? and if they were from the single sample per condition, was the experiment replicated is necessary. I believe that becomes important for the reader to decide the strength of the protein level result. 

Reviewer 3 Report

Thank you for the responses from the authors. The revision addressed the majority of my concerns, especially on the EMT part. 

I have several minor suggestions which may strengthen the manuscript:

1. Please have a discussion on other novel natural plant extraction in inhibiting pancreatic cancer, especially the ones focusing on pancreatic cancer metastasis and invasion. e.g. Pao Pereira extract derived from Geissospermum vellosii (https://www.mskcc.org/cancer-care/integrative-medicine/herbs/pao-pereira). 

2. Please have a discussion on the 1. Limitation of the study (lack of animal study, only used an ascites derived cell lines); 2. Potential mechanism of the synergistic effect of combination of KV and oxaliplatin in inhibiting PDAC. ABCG transporter? Inhibiting pancreatic cancer stem cells?
